# MASKCAPTIONER: LEARNING TO JOINTLY SEGMENT AND CAPTION OBJECT TRAJECTORIES IN VIDEOS

## ABSTRACT

Dense Video Object Captioning (DVOC) is the task of jointly detecting, tracking, and captioning object trajectories in a video, requiring the ability to understand spatio-temporal details and describe them in natural language. Due to the complexity of the task and the high cost associated with manual annotation, previous approaches resort to disjoint training strategies, potentially leading to suboptimal performance. To circumvent this issue, we propose to generate captions about spatio-temporally localized entities leveraging a state-of-the-art VLM. By extending the LVIS and LV-VIS datasets with our synthetic captions (LVISCap and LV-VISCap), we train MaskCaptioner, an end-to-end model capable of jointly detecting, segmenting, tracking and captioning object trajectories. Moreover, with pretraining on LVISCap and LV-VISCap, MaskCaptioner achieves state-of-the-art DVOC results on three existing benchmarks, VidSTG, VLN and BenSMOT.

## 1 INTRODUCTION

A fundamental aim of computer vision is to enable machines to understand videos with human-like acuity in perceiving and reasoning about the world. Recent advances have led to remarkable progress in both spatio-temporal localization (Ren et al., 2015; Redmon et al., 2016; Carion et al., 2020; He et al., 2017; Wojke et al., 2017; Zhang et al., 2022b) and vision-language understanding (Vinyals et al., 2015; Sun et al., 2019; Yu et al., 2019; Yang et al., 2021; Chen et al., 2020; Jia et al., 2021; Zellers et al., 2021; Bain et al., 2021; Lu et al., 2019; Kamath et al., 2021). However, building vision-language models that can simultaneously reason about spatially localized objects and temporal dynamics of a complex scene remains a significant challenge, motivated by many real-world applications including autonomous driving (Kim et al., 2019; Atakishiyev et al., 2024), human-computer interaction (Shridhar et al., 2020; Ahn et al., 2022), or video editing (Molad et al., 2023; Jeong and Ye, 2023). Dense Video Object Captioning (DVOC) (Zhou et al., 2023) serves as a key benchmark for this purpose, as it requires to jointly localize, track, and describe in natural language *all* visual entities in a video.

Manual annotation for such a fine-grained task is particularly expensive, leading to a scarcity of datasets with densely annotated object-level video descriptions. To tackle DVOC, prior work resorted to alternative training approaches: Zhou et al. (2023) propose a disjoint training strategy, decomposing the problem into subtasks and training a model sequentially on datasets for each subtask. Choudhuri et al. (2024) leverage the pretraining of multiple specialized models to alleviate the need for object-level annotations. Both methods allowed to perform DVOC while circumventing the need for costly annotations, but the lack of end-to-end training with densely annotated object-level supervision may lead to suboptimal performance.

We propose to address this limitation by generating synthetic object-level annotations, motivated by the recent success of LLM-generated supervision (Liu et al., 2023; Abdin et al., 2024) and the growing visual capacities of Vision Language Models (VLMs) (Alayrac et al., 2022; Li et al., 2022; 2023a; Team et al., 2023; 2024; Achiam et al., 2023; Grattafiori et al., 2024; Bai et al., 2023). To the best of our knowledge, our work is the first to generate localized, object-level captions for DVOC. To this end, we introduce a multi-modal prompting strategy leveraging a state-of-the-art VLM, and extend two segmentation datasets, LVIS (Gupta et al., 2019b) for images and LV-VIS (Wang et al., 2023) for videos, to be the first DVOC training sets with (mask, box, category, caption) annotations for all objects, dubbed LVISCap and LV-VISCap, see figure 1.

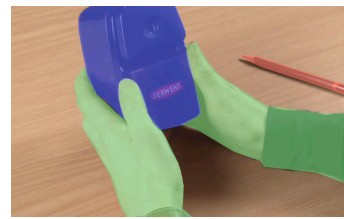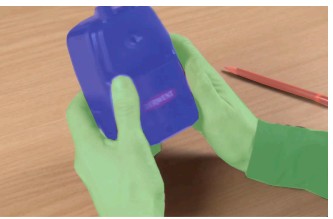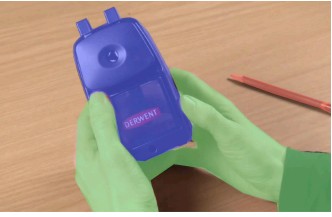

A person is holding a pencil sharpener in their hands.
The black Derwent pencil sharpener is being held and rotated in a person's hand.
A graphic pencil with a red end sits on a wooden surface next to a pencil sharpener.

Figure 1: Examples of synthetic captions in our LV-VISCap dataset.

Using our generated datasets, we extend the DVOC task, traditionally formulated as detection (Zhou et al., 2023), to segmentation and train MaskCaptioner, the first end-to-end model that can jointly produce (mask, caption) pairs for all object trajectories in a video. We show that (i) our generated datasets, LVISCap and LV-VISCap, largely benefit MaskCaptioner's DVOC performance, (ii) our MaskCaptioner outperforms previous state-of-the-art models on the VidSTG, VLN and BenSMOT DVOC benchmarks and (iii) we can extend the DVOC task to segmentation.

Overall, our contributions can be summarized as follows:

1. We introduce a VLM-based method to generate synthetic object captions for videos, and extend the LVIS and LV-VIS datasets to be the first unified DVOC training set with object captions, boxes, and segmentation masks: LVISCap and LV-VISCap.
2. Using our unified generated data, we train MaskCaptioner, the first end-to-end model to jointly detect, segment, track and caption objects in a video.
3. MaskCaptioner achieves state-of-the-art DVOC results on the three existing benchmarks : VidSTG, VLN and BenSMOT.

The code and generated annotations will be made publicly available online.

## 2 RELATED WORK

**Open-Vocabulary Video Instance Segmentation (OV-VIS).** The OV-VIS task aims to segment, track, and classify objects from an open set of categories in videos (Guo et al., 2025; Wang et al., 2023), using datasets such as LV-VIS (Wang et al., 2023). State-of-the-art methods (Guo et al., 2025; Wang et al., 2023; Fang et al., 2025) commonly use query-based approaches that classify objects by matching visual features with CLIP embeddings (Radford et al., 2021). Methods like OVFormer (Fang et al., 2025) or BriVIS (Cheng et al., 2024) improve this approach by better aligning visual queries with the CLIP space. Unlike these methods focused on CLIP feature matching for classification, our work explores describing objects in natural language (Li et al., 2023a).

**Localized vision-language understanding.** Going beyond pioneering vision-language tasks such as visual question answering (Antol et al., 2015) or image captioning (Chen et al., 2015), recent work has explored spatial understanding tasks that require localizing natural language queries in images. This includes referred expression segmentation (Kazemzadeh et al., 2014; Yu et al., 2018; Yang et al., 2022b), image grounding (Rohrbach et al., 2016; Plummer et al., 2015), reasoning segmentation (Lai et al., 2024; Wang and Ke, 2024), spatio-temporal video grounding (Zhang et al., 2020; Yang et al., 2022a) and grounded visual question answering (Zhu et al., 2016; Xiao et al., 2024; Lei et al., 2018; 2019). While these tasks typically require localizing one or a few entities, dense captioning (Johnson et al., 2016; Wu et al., 2024) aims to spatially localize and describe in natural language *all* salient regions in images. Our work addresses the more challenging task of predicting *both* object trajectories and descriptions for *all* objects in a video.

**Dense Video Object Captioning (DVOC).** The DVOC task aims at jointly detecting, tracking, and describing the trajectory of all visual entities in a video. DVOC-DS (Zhou et al., 2023) tackles this task by generating frame-wise object box proposals (Zhou et al., 2019; Cai and Vasconcelos, 2018) that are tracked (Zhou et al., 2022) before feeding aggregated and cropped features to a generative image-to-text decoder (Wang et al., 2022). To cope with the lack of DVOC annotations, the model is trained disjointly on various subtasks: object detection using COCO (Lin et al., 2014), image object-level captioning using Visual Genome (Krishna et al., 2017), video scene-level captioning using SMiT (Monfort et al., 2021) and video object tracking using AugCOCO (Lin et al., 2014).

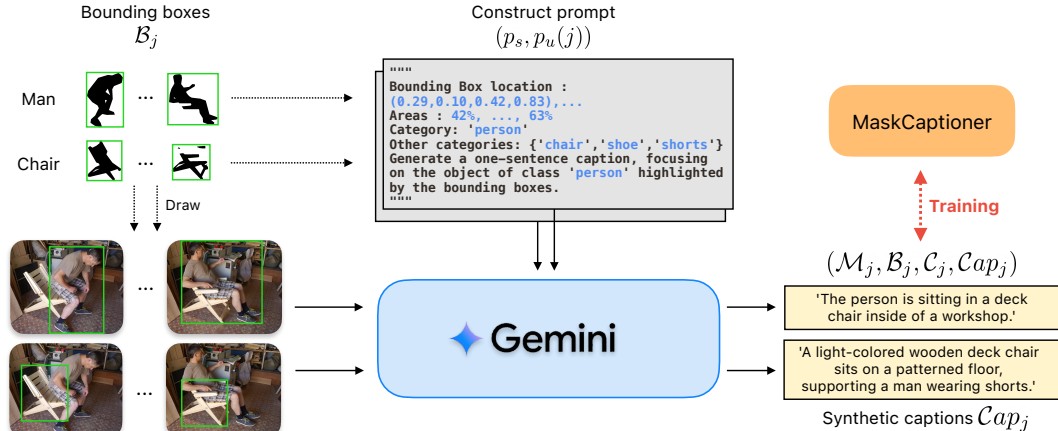

Figure 2: **Our MaskCaptioner data annotation pipeline**: For each object, we extract its bounding boxes $\mathcal{B}_j$ from its annotated masks $\mathcal{M}_j$, and draw them in the video. The video with drawn bounding boxes $\hat{x}_j$, along with a prompt $(p_s, p_u(j))$ including additional information such as the label of the object to caption $\mathcal{C}_j$, the labels and bounding boxes of other objects in the image, is fed to a state-of-the-art VLM (Gemini 2.0 Flash (Team et al., 2023)) to generate the object-level caption. $p_s$ denotes the static prompt with general instructions and $p_u(j)$ the dynamic prompt with annotation information. We use this pipeline to generate the LVISCap and LV-VISCap datasets, used to train MaskCaptioner.

OW-VisCapTor (Choudhuri et al., 2024) extends the DVOC task to segmentation masks. Its architecture relies on an object abstractor using a prompt encoder and transformer blocks, an inter-query contrastive loss to ensure object queries are diverse, and an object-to-text abstractor that connects these queries to a frozen LLM, generating rich, object-centric captions for each detected instance. While proposing to extend DVOC to segmentation, OW-VisCapTor is hindered by the absence of paired (mask, caption) annotations, hence segmentation and DVOC are tackled in isolation using separate models. Closely related, SMOTer (Li et al., 2024) extends multi-object tracking to object-level captioning, video-level captioning and object-relation predictions, and introduce a hand-annotated dataset, BenSMOT. However, BenSMOT focuses on humans only, whereas the standard DVOC task considers all visual entities in the video. In this work, we automatically generate DVOC datasets, LVISCap and LV-VISCap, and train MaskCaptioner, a model that can end-to-end detect, segment, track and caption object trajectories in videos.

**Vision-language data generation.** A promising approach for intricate vision-language tasks is to generate visual annotations using Large Language Models (LLMs) or Vision Language Models (VLMs). LLaVA (Liu et al., 2023) leverages a LLM to generate conversations and detailed descriptions from paired image-text annotations (Chen et al., 2015). This approach has been followed to generate large-scale instruction tuning data for video understanding (Maaz et al., 2023; Li et al., 2023b). Recent research has focused on the generation of spatially grounded text-image data using LLMs/VLMs: Shikra (Chen et al., 2023) generates grounded QA pairs from Flickr30K Entities (Plummer et al., 2015), LLaVA-Grounding (Zhang et al., 2024) and GLaMM (Rasheed et al., 2024) generate grounded conversational data from datasets such as COCO (Lin et al., 2014). Recently, GROVE (Kazakos et al., 2025) extended GLaMM to generate grounded, temporally consistent video captions, extending dense captioning to the video domain. These methods predominantly operate at the scene level and do not produce localized, object-level video descriptions. In contrast, we introduce a multi-modal prompting strategy to leverage VLMs into generating fine-grained captions for individual object trajectories across time. Recently, Yuan et al. (2025) proposed a related VLM-based approach to generate object-level video captions designed for long sentences referring segmentation. Differently, we focus on the task of Dense Video Object Captioning where no text prompt is given to the model.

## 3 METHOD

Dense Video Object Captioning (DVOC) (Zhou et al., 2023) is the task of jointly detecting, tracking and captioning objects trajectories in a video, i.e. producing a (bounding box or mask, caption) pair for each object in a video at each timestamp they appear. Such densely annotated data is lacking for video, as there is currently no available training dataset including captions for all object trajectories in

the videos. The spatio-temporal grounding dataset VidSTG (Zhang et al., 2020) has been repurposed to DVOC but only includes annotations for few objects per video and a limited number of frames per video.

We address the lack of data by introducing a strategy for synthetic DVOC data generation, allowing unified training of a DVOC model on (trajectory, caption) pairs for each object, as presented in Section 3.1. With this strategy, we extend the LV-VIS dataset (Wang et al., 2023), which includes both boxes and masks in their (trajectory, label) annotations for all objects, to a variant, dubbed LV-VISCap, which additionally includes synthetically generated captions for each trajectory. To enable end-to-end training on this rich data using segmentation masks, we build an architecture based on a state-of-the-art Open-Vocabulary Video Instance Segmentation (OV-VIS) model OVFormer (Fang et al., 2025), as described in Section 3.2.1. As OVFormer is designed for classification, we extend it with a captioning head (Choudhuri et al., 2024), as explained in Section 3.2.2. Finally, we present in Section 3.3 the losses used to train our model.

### 3.1 DVOC DATA GENERATION

We start from the LV-VIS dataset (Wang et al., 2023) which contains (segmentation masks, category) manual annotations for all objects and timestamps of the videos. To automatically collect DVOC data, one challenge is to generate accurate object-level captions for each trajectory. For this, we leverage a state-of-the-art VLM (Gemini 2.0 Flash (Team et al., 2023)) and feed it with videos where the object to caption is marked with drawn bounding boxes, as illustrated in Figure 2.

**Visual Prompt.** Formally, let $x \in \mathbb{R}^{N \times H \times W \times 3}$ be a video clip of length $N$, with associated mask and category annotations for $M$ objects in the video: $(\mathcal{M}_j, \mathcal{C}at_j), \ j \in 1, ..M$. We first extract bounding boxes annotations from the ground-truth masks, $\mathcal{B}_j \in \mathbb{R}^{N \times 4}, \ j \in 1, ..., M$. We draw each object boxes on a separate copy of the video, and denote as $a \wedge b$ the operation of drawing box $b$ on frame $a$. We obtain a visual prompt $\hat{x}_j^i$ for each object trajectory: $\hat{x}_j^i = x^i \wedge \mathcal{B}_j$ for $i \in 1, ..., N, \ j \in 1, ..., M$

Note that in practice, we subsample $N$ to 4 uniformly sampled video frames as we found it produces representative enough visual content for the captioning task.

**Text prompts.** In detail, the prompt we feed the VLM is composed of the previously described visual prompt $\hat{x}_j$, a system prompt $p_s$ and a user prompt $p_u(j)$. The system prompt is static for all objects/videos and gives gen-

Table 1: **Impact of the prompting strategy on caption quality**. Scores are given by an expert human evaluator from 0 to 2 (incorrect, partially correct, or correct) on a subset from the LV-VIS validation set, and brought to 0-100 range. For the mask visual prompt experiments, we use our best prompt with either the object's bounding boxes or center point coordinates as a localization cue in the text prompt.

| Visual prompt | Prompting method | Average rating |
|---|---|---|
| bounding boxes | single frame | 26.8 |
| | + multiple frames | 27.1 |
| | + detailed instructions | 29.5 |
| | + category labels | 80.7 |
| | + bbox coordinates | 83.1 |
| | + bbox area | 84.3 |
| | + few shot examples | **85.1** |
| mask boundaries | center point coordinates | 75.9 |
| | bbox coordinates | 77.1 |

eral instructions (e.g. "Generate a caption about a queried object highlighted with bounding boxes."), rules (e.g. "Do not mention the bounding boxes in the caption."), format, and an example. The user prompt $p_u(j)$ however, is constructed for each annotation and enriched with informations about the specific query to help the model. The user prompt encodes textual bounding box coordinates, area, category name of the queried object, and category names from other objects in the image. Passing information through different channels (visual prompt, text prompt) helps the model focusing on the queried object and being accurate when describing the scene. Also, this complementary information can lead the model to reason about the objects (e.g. area being small for an object of category 'elephant' implies the object is most likely part of the background). The different prompting cues have been ablated in Table 1. In particular, showing that including textual semantic and localization cues in the prompt helps the model to focus on the queried object and generate more accurate object-focused captions. We notice that using the segmentation masks as the visual prompt for the model results in less accurate object captions, which might result from a poorer alignment with the localization cues included in the text prompt. The full template details for the model prompt construction and a more detailed ablation are given in Appendix A.4.1 and A.2.

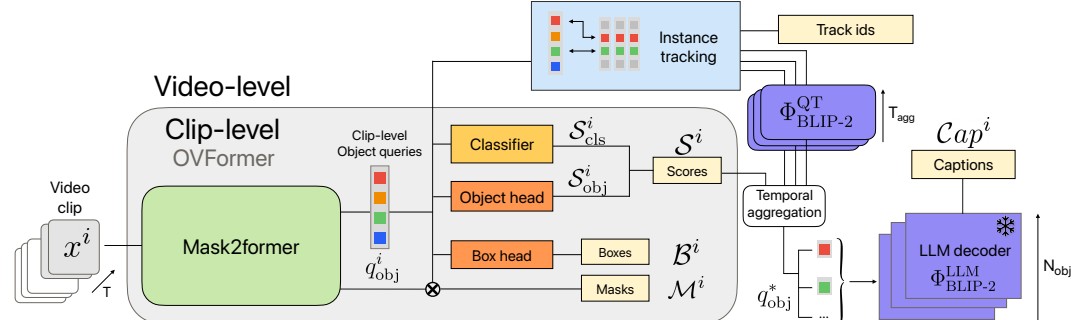

Figure 3: **Our MaskCaptioner architecture** jointly segments and captions objects in videos. For each clip of $T$ frames, we obtain $N_{\text{obj}}$ clip-level object queries through Mask2Former (Cheng et al., 2021), and yield associated score, mask and box predictions. At a video-level, we match the predicted object queries with the previously processed clips with the Hungarian bipartite matching algorithm, and perform instance tracking (Zhu et al., 2024). For each track, we sample $T_{\text{agg}}$ clips uniformly across the video, and aggregate the tracked queries to obtain a single video query per object, which we can feed to the LLM captioning head (Li et al., 2023a), producing a single caption per trajectory.

**The LVISCap and LV-VISCap datasets.** For each object $j$ in the video, we prompt Gemini 2.0 Flash (Team et al., 2023) with the visual and textual prompts ($\hat{x}_j$ , $p_s$ and $p_u(j)$) to output a synthetic caption $\mathcal{Cap}_j$. The full DVOC annotation is $(\mathcal{M}_j, \mathcal{B}_j, \mathcal{Cat}_j, \mathcal{Cap}_j)$. We repeat this process for each object in each videos of LV-VIS, generating over 19.5k synthetic captions over 3.9k videos covering $1,020$ object classes, and with an average length of 13.9 words. We repeat this process with the LVIS dataset to support image pre-training, applying the same method as above described with each image considered a video of len $N = 1$, and obtain our DVOC training sets: LVISCap and LV-VISCap.

## 3.2 MASKCAPTIONER ARCHITECTURE

To enable end-to-end training on the previously described DVOC data including segmentation masks, our architecture, illustrated in Figure 3, processes videos as $N_{\text{clip}}$ clips of $T$ consecutive frames each, and is composed with: (i) at the clip level, an instance segmentation and detection component, see Section 3.2.1 (ii) at the video-level, a tracking module and a captioning head, see Section 3.2.2.

### 3.2.1 INSTANCE SEGMENTATION AND DETECTION

**Background: OVFormer (Fang et al., 2025).** Our clip-level instance segmentation component is based on OVFormer (Fang et al., 2025), which we shortly describe next. On a high level, OVFormer augments Mask2Former (Cheng et al., 2021) with a classification head to handle Open-Vocabulary Video Instance Segmentation. Mask2Former learns clip-level object queries with a transformer decoder which cross-attends to the features extracted with a visual backbone and are refined with a pixel decoder. Formally, each clip $x^i_{\text{clip}} \in \mathbb{R}^{T \times 3 \times H \times W}$ is composed of $T$ consecutive frames for images of resolution $(H,W)$, and processed independently by the OVFormer model which outputs clip-level object queries $q^i_{\text{obj}}$, associated mask predictions $\mathcal{M}^i$, classification and objectness scores $\mathcal{S}^i_{\text{cls}}$ and $\mathcal{S}^i_{\text{obj}}$: $(q_{\text{obj}}, \mathcal{M}^i, \mathcal{S}^i_{\text{cls}}, \mathcal{S}^i_{\text{obj}}) = \Phi_{\text{OVFormer}}(x^i)$ where $q^i_{\text{obj}} \in \mathbb{R}^{N_{\text{obj}} \times D}$, $\mathcal{M}^i \in \mathbb{R}^{N_{\text{obj}} \times T \times H \times W}$, $S^i_{\text{cls}} \in \mathbb{R}^{N_{\text{clip}} \times N_{\text{obj}} \times N_{\text{cls}}}$ and $S^i_{\text{obj}} \in \mathbb{R}^{N_{\text{clip}} \times N_{\text{obj}}}$.

**Detection head.** We extend this segmentation module with detection by using a 4-layer MLP to generate boxes on top of the object queries $q^i_{\text{obj}}$: $\mathcal{B}^i = \text{BoxHead}(q^i_{\text{obj}}) \in \mathbb{R}^{N_{\text{obj}} \times T \times 4}$.

**Confidence scores.** Note that OVFormer (Fang et al., 2025) only computes class-aware query-wise confidence scores over the full video. However, for objects appearing only in a small subset of frames in the video this strategy could result in inaccurate scores. Moreover, for DVOC, we wish to avoid redundant predictions i.e. having two queries predicting a similar trajectory. Hence we additionally compute *class-agnostic* query-wise confidence scores *for each clip* $\mathcal{S}^i_{\text{cls}*}$, by taking the maximum classification score over all labels $c \in 1, ..., N_{\text{cls}}$ for each query and clip:

$\mathcal{S}^i_{\text{cls}*} = \max_c(S^i_{\text{cls}}(c)) \in \mathbb{R}^{N_{\text{clip}} \times N_{\text{obj}}}$. Finally we derive the per-clip score $\mathcal{S}^i = \sqrt{\mathbf{S}^i_{\text{cls}*} \times \mathbf{S}^i_{\text{obj}}}$ which we use for filtering predictions below a threshold $t_{\text{thresh}}$ at inference-time for every time step.

### 3.2.2 INSTANCE TRACKING AND CAPTIONING

**Tracking module.** To derive the output video-level trajectories from the clip-level predictions, we need to obtain a matching between the queries at time $i$ and the queries at time $i + 1$. For this, we perform tracking between the clips using the top-K enhanced query-matching module from Zhu et al. (2024). For each clip, this module keeps a memory bank containing the queries from the $T_{\text{match}}$ previous clips. Among these, it identifies the $K_{\text{match}}$ most matched clips, and computes the optimal assignment using the Hungarian bipartite matching algorithm. Using the $K_{\text{match}}$ most matched clips helps reducing error propagation compared to the OVFormer (Fang et al., 2025) tracking module, which maps queries from time $i + 1$ to time $i$ directly. Notably, we can keep track of objects that disappear and re-appear in a video, whereas they are automatically lost using the OVFormer tracking module.

This method is referred to as semi-online tracking as we represent objects at a clip-level and associate between the clips in an online fashion. This offers the advantages of being flexible (fully-online for clips of length $T = 1$) and to arbitrate between using multi-frame information and memory constraints for long videos.

**Captioning head.** To caption tracked object trajectories, we adapt the captioning head from Choudhuri et al. (2024) based on BLIP-2 (Li et al., 2023a). The BLIP-2 decoder processes object queries one by one using masked-attention conditioned with the predicted masks, before projecting the resulting object query into the LLM space for caption prediction. However, for consistency and efficiency, we predict a single video-level caption per tracked object query, replacing clip-level prediction.

**Query aggregation.** Let's consider $\Phi_{\text{BLIP2}}^{\text{QT}}$ the BLIP-2 query transformer, $\Phi_{\text{BLIP2}}^{\text{LLM}}$ the LLM decoder, $q_{\text{obj}}^i(j) \in \mathbb{R}^{1 \times D}$ and $\mathcal{S}_{\text{obj}}^i(j) \in \mathbb{R}$ respectively the query and the detection score for object $j$ from clip $i$. For each object, we aggregate the tracked queries over time after they are processed by the BLIP-2 query transformer, by sampling a set $\mathcal{I}_{\text{agg}}$ of $T_{\text{agg}}$ clips uniformly across the video. We obtain a video query for each object $j$: $q_{\text{obj}}^*(j) = \sum_{i \in \mathcal{I}_{\text{agg}}} \mathcal{S}^i(j) \times \Phi_{\text{BLIP-2}}^{\text{QT}}(q_{\text{obj}}^i(j), \mathcal{M}_j^i)$. We can compute the video captioning prediction for the tracked object: $\mathcal{C}ap(j) = \Phi_{\text{BLIP2}}^{\text{LLM}}(q_{\text{obj}}^*(j))$ for $j \in 1, ...N$.

### 3.3 MODEL TRAINING

We train MaskCaptioner with a combination of clip-level and video-level losses. For each clip, we predict masks/boxes, classification scores and captions, and derive the clip-level training objective as the following combination of supervised losses:

$$\mathcal{L}_{\text{clip-level}} = \mathcal{L}_{\text{seg}} + \mathcal{L}_{\text{det}} + \mathcal{L}_{\text{s}} + \mathcal{L}_{\text{cap}} \tag{1}$$

where $\mathcal{L}_{\text{seg}}$ and $\mathcal{L}_{\text{s}}$ are the VIS losses from Fang et al. (2025), i.e. $\mathcal{L}_{\text{seg}} = \lambda_{\text{dice}}\mathcal{L}_{\text{dice}} + \lambda_{\text{ce}}\mathcal{L}_{\text{ce}}$, with $\mathcal{L}_{\text{dice}}$ and $\mathcal{L}_{\text{ce}}$ the dice and cross-entropy segmentation losses respectively, and $\mathcal{L}_{\text{s}} = \lambda_{\text{cls}}\mathcal{L}_{\text{cls}} + \lambda_{\text{obj}}\mathcal{L}_{\text{obj}}$ where $\mathcal{L}_{\text{cls}}$ and $\mathcal{L}_{\text{obj}}$ are the cross-entropy losses for classification and objectness. We add detection and captioning losses $\mathcal{L}_{\text{det}} = \lambda_{l_1}\mathcal{L}_{l_1} + \lambda_{\text{giou}}\mathcal{L}_{\text{giou}}$ where $\mathcal{L}_{l_1}$ and $\mathcal{L}_{\text{giou}}$ are detection losses from Yang et al. (2022a), and $\mathcal{L}_{\text{cap}} = \lambda_{\text{clip-lm}}\mathcal{L}_{\text{lm}}$, with $\mathcal{L}_{\text{lm}}$ the cross-entropy language modeling loss (Zhou et al., 2023).

When including the temporal aggregation module for captioning, we train the captioning head at the video level, i.e. we predict a caption per object for the full video after the tracking is performed and each object-query has been augmented across time.

$$\mathcal{L}_{\text{video-level}} = \lambda_{\text{vid-lm}}\mathcal{L}_{\text{lm}} \tag{2}$$

MaskCaptioner can be trained in a completely end-to-end manner. However, in practice, we train the model in two-stages for most of the experiments to alleviate memory constraints: we first train the segmentation/detection and classification model, then freeze it and tune the captioning head. The captioning head is trained either at the clip-level, or at the video-level when enabling the temporal aggregation module (i.e. $\lambda_{\text{clip-lm}} = 0$ or $\lambda_{\text{vid-lm}} = 0$). For each loss when they are computed we set their respective weights to $\lambda_{\text{dice}}, \lambda_{\text{ce}}, \lambda_{l_1} = 5, \lambda_{\text{giou}}, \lambda_{\text{cls}}, \lambda_{\text{obj}} = 2$, and ($\lambda_{\text{clip-lm}} = 1$ or $\lambda_{\text{vid-lm}} = 1$).

# 4 EXPERIMENTS

## 4.1 EXPERIMENTAL SETTING

**Datasets.** **LVIS** (Gupta et al., 2019a) and **LV-VIS**(Wang et al., 2023) are large-vocabulary instance segmentation datasets, respectively for image and video. **LVISCap** and **LV-VISCap** denote our extensions of LVIS and LV-VIS (see Section 3.1), with respectively $1,2M/244k$ synthetic image object captions for the training/validation of LVISCap and $16k/3.7k$ synthetic video object captions for LV-VISCap (average of $5.4$ objects per video). Note that in the absence of annotations on the test sets of LVIS and LV-VIS, we only extend the training and validation sets with captions, and use the validation set for evaluation.

**VidSTG**(Zhang et al., 2020) is a spatio-temporal video grounding dataset containing text descriptions serving as queries, which Zhou et al. (2023) propose to use for DVOC evaluation. The repurposed training and validation sets count $5.4k/602$ videos for $15.1k/1.6k$ object trajectories with captions respectively. **Video Localized Narratives (VLN)** is similarly repurposed (Zhou et al., 2023). For each of the $5.1k$ training and $2.4k$ validation videos, the dataset contains 3 sparsely annotated frames with non exhaustive captions. **BenSMOT** contains bounding box trajectories and associated captions focusing exclusively on humans in videos, with an average of 2.2 instances per video. It counts $2.2k$ videos for training and $1k$ for evaluation. More details about the datasets are given in Appendix A.4.2.

**Evaluation Metrics**. Following prior work, we evaluate DVOC using the CHOTA metric introduced by Zhou et al. (2023), which extends the widely used multi-object tracking HOTA metric (Luiten et al., 2021). CHOTA decomposes the DVOC task into three components: detection accuracy (DetA)(Luiten et al., 2021), association accuracy (AssA)(Luiten et al., 2021), and captioning accuracy (CapA)(Zhou et al., 2023). These components reflect the model's ability to (i) correctly localize objects, (ii) maintain their identity across frames, and (iii) generate accurate natural language descriptions. CHOTA is defined as the geometric mean of the three components: $\text{CHOTA} = \sqrt[3]{\text{DetA} \cdot \text{AssA} \cdot \text{CapA}}$.

Matching between predicted and ground-truth trajectories is performed using Intersection-over-Union (IoU) thresholds, similarly to standard tracking evaluations (Milan et al., 2016). To extend the metric to segmentation masks, we simply replace box-based IoU with mask-based IoU when computing CHOTA with segmentation masks.

**Implementation details.** Following OVFormer (Fang et al., 2025) we used ResNet50 (He et al., 2016) and SwinBase (Liu et al., 2021) visual backbones. Our Mask2former (Cheng et al., 2022) transformer decoder has 11 layers, and our captioning head is based on the BLIP-2 (Li et al., 2023a) decoder with OPT-2.7B LLM (Zhang et al., 2022a). For LV-VIS experiments we tune the model end-to-end with clip-level supervision only. For other experiments, we first train the segmentation/detection model, then freeze it and tune the captioning head. For VidSTG and BenSMOT experiments we use video-level tuning for captioning with temporal aggregation. For VLN experiments we do not use it since the videos are very short. For the largest dataset (COCO + LVIS) the optimization takes 2 days on 4 H100 GPUs. More implementation details and hyper-parameters for different experiments are given in Appendix A.4.3.

## 4.2 RESULTS

### 4.2.1 BENEFITS OF TRAINING ON LVISCAP AND LV-VISCAP

We first study the impact of integrating our generated datasets for DVOC training in Table 2. We train MaskCaptioner on our synthetic video set, LV-VISCap, and progressively add LVISCap image pretraining. Results are reported on the LV-VISCap validation set. First, we observe that MaskCaptioner achieves strong DVOC results with training only on LVISCap or LV-VISCap, demonstrating the effectiveness of our architecture. Importantly, combining both LVISCap and LV-VISCap leads to best results, showing the benefit of both our generated datasets. Moreover, we observe that our MaskCaptioner is robust to the choice of the visual backbone. Note that the AssA score depends on the detections, hence a worse DetA score with lower recall but higher precision can make the tracking easier: this explains the higher AssA score from the model without LV-VIScap tuning. In Fig. 4, we show that the CapA performance is logarithmically correlated with the quantity of training captions, suggesting that generating more data with our approach might bring further improvements.

Table 2: **Impact of training with LVISCap and LV-VISCap and of the visual backbone on LV-VISCap DVOC.**

| Backbone | LVIScap | LV-VIScap | CapA | DetA | AssA | CHOTA |
|---|---|---|---|---|---|---|
| **SwinB** | - | ✓ | 37.9 | 48.1 | 89.5 | 54.7 |
| | ✓ | - | 30.0 | 34.3 | **93.2** | 45.8 |
| | ✓ | ✓ | **43.6** | **54.3** | 89.1 | **59.5** |
| **ResNet50** | ✓ | ✓ | 39.0 | 51.1 | 88.5 | 56.1 |

### 4.2.2 COMPARISON WITH THE STATE OF THE ART

We compare MaskCaptioner to state-of-the-art DVOC methods following the standard evaluation protocol on three existing benchmarks : VidSTG in Table 3, VLN in Table 4 and BenSMOT in Table 5. DVOC-DS (Zhou et al., 2023) reports results without pretraining, and with their disjoint training training strategy, while OW-VISCaptor (Choudhuri et al., 2024) leverages Mask2former pretrained on COCO for instance segmentation. In Table 3, we include results with the same pretraining data as these methods and show the impact of using our data instead. Pretraining MaskCaptioner on COCO yields better detection and tracking compared to OW-VISCaptor but comparable captioning performance due to the similar captioning head design. Including our data

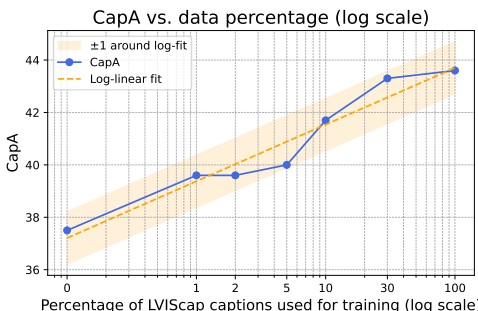

Figure 4: **Impact of the generated data scale on the CapA metric.** We train MaskCaptioner on a varying percentage of LVISCap captions and finetune on LV-VISCap.

leads to an improvement on all metrics and especially captioning with a 6.7 CapA increase (15% relative improvement). When pretraining on the disjoint DVOC-DS set, we observe a substantial gain in the captioning metric due to the model design. Moreover, we show that using our pretraining sets results in a further performance improvement, while additionally allowing a unified, much faster training (2032 GPU hours (Zhou et al., 2023) vs 208 for our approach). Eventually, our proposed approach can output segmentation masks unlike other methods.

Table 3: **Comparison with the state of the art on VidSTG DVOC validation set.** All models are finetuned on VidSTG. "temp. agg." refers to including the query temporal aggregation module.

| Method | Pretraining set | CapA | DetA | AssA | CHOTA |
|---|---|---|---|---|---|
| OW-VISCaptor (Choudhuri et al., 2024) | COCO | 43.9 | 60.1 | 54.0 | 53.0 |
| MaskCaptioner (ours) | COCO | 44.3 | 65.1 | 70.2 | 58.7 |
| DVOC-DS (Zhou et al., 2023) | COCO + VG + SMIT + AugCOCO | 39.7 | 65.8 | 70.4 | 56.9 |
| MaskCaptioner (ours) | COCO + VG + SMIT + AugCOCO | 50.1 | 65.0 | 69.2 | 60.9 |
| MaskCaptioner (ours) | COCO + LVIScap + LV-VIScap | 51.0 | 66.8 | 71.0 | 62.3 |
| + temp agg | COCO + LVIScap + LV-VIScap | **52.7** | **66.8** | **71.0** | **63.0** |

Table 4: **Comparison with state of the art on the VLN DVOC validation set.** Mask loss refers to using the segmentation masks in the detection loss. All models are finetuned on VLN.

| Method | Mask loss | CapA | DetA | AssA | CHOTA |
|---|---|---|---|---|---|
| DVOC-DS Zhou et al. (2023) | - | 17.7 | 44.3 | 89.5 | 41.3 |
| MaskCaptioner (ours) | - | 21.4 | 48.7 | 89.7 | 45.4 |
| MaskCaptioner (ours) | ✓ | **22.9** | **50.1** | **92.7** | **47.4** |

Table 5: **Comparison on the BenSMOT validation set.** CapA, and thus CHOTA are not reported on this dataset. All models are finetuned on BenSMOT.

| Method | DetA | AssA | CIDEr |
|---|---|---|---|
| SMOTer (Li et al., 2024) | 80.8 | 73.7 | 8.7 |
| DVOC-DS Zhou et al. (2023) | 90.8 | **89.6** | 25.4 |
| MaskCaptioner (ours) | **91.6** | 87.4 | 39.9 |
| + temp agg | **91.6** | 87.4 | **40.1** |

We further evaluate MaskCaptioner on VLN in Table 4 and BensMOT in Table 5. On both benchmarks, our approach improves the detection while the tracking remains competitive. Most important, the captioning metrics improve by a large margin (+5.2 CapA on VLN, +14.7 CIDEr on BenSMOT

Table 6: **Automatic vs manual annotations for evaluation on a subset from LV-VIScap validation** with 50 videos and 233 objects trajectories ; "**automatic**" and "**manual**" annotations stands for synthetic or human annotated captions. All models are trained on LVISCap and tuned on our LV-VISCap training set.

| Annotation type | LVIScap captions | CapA | DetA | AssA | CHOTA |
|---|---|---|---|---|---|
| **automatic** | - | 33.0 | 48.4 | 89.6 | 52.3 |
| | ✓ | **40.5** | **50.0** | **91.0** | **56.7** |
| **manual** | - | 22.5 | 48.4 | 89.6 | 46.0 |
| | ✓ | **33.2** | **50.0** | **91.0** | **53.1** |

Table 7: **Impact of the inference clip on LV-VIScap.** All models are trained on LVIScap then LV-VIScap training set.

| Clip length | mAP |
|---|---|
| 3 | 26.0 |
| 4 | 26.1 |
| 5 | 26.6 |
| 6 | 26.3 |
| 7 | **26.8** |

compared to state of the art). Additionally, our method is able to jointly segment objects, which further improves the performance as shown in Table 4.

### 4.2.3 ADDITIONAL ABLATIONS

**Annotation bias.** To evaluate the bias introduced by evaluating on LV-VISCap synthetic captions (see Table 2), we annotate a representative subset of our LV-VISCap datasets by hand and compare the impact of our LVISCap captions when evaluating MaskCaptioner on *automatic* vs *manual* data. Our subset comports 50 videos from the LV-VISCap validation set with 233 object trajectories annotated by hand. The results are presented in Table 6. We observe that both the evaluation on automatic and manual data show a comparable improvement on the CapA and CHOTA metric when using our LVISCap captions for the training. This result confirms the importance of our synthetic captions for DVOC performance, and further shows that the bias introduced by evaluating on synthetic LV-VISCap captions is marginal.

Table 8: **Impact of temporal aggregation on VidSTG validation** (with finetuning). All methods are pretrained on COCO + LVIScap + LV-VIScap. Multi-clip results are obtained with weighted mean temporal aggregation, except for † which uses arithmetic mean.

| num clips | clip selection | CapA | CHOTA |
|---|---|---|---|
| 1 | best score | 51.0 | 62.3 |
| 1 | middle frame | 46.9 | 60.6 |
| 4 | uniform† | 49.1 | 61.5 |
| 4 | uniform | 51.6 | 62.6 |
| 8 | uniform | 52.7 | 63.0 |
| 16 | uniform | 53.8 | 63.4 |
| 32 | uniform | **55.4** | **64.0** |

**Clip length.** We show in Table 7 the impact of the clip length used for the MaskCaptioner inference. A higher clip length leads to temporally richer queries, improving detection and tracking. Note that in our implementation, we follow OVFormer (Fang et al., 2025) and use a clip length of $T = 5$.

**Temporal aggregation.** We show the impact of temporal aggregation on DVOC performance in Table 8. Using more clips to aggregate consistently improves the captioning scores, and weighting the clips with the detection scores helps the captioning. We also observe that performing captioning based on the single clip with best score performs relatively well, which highlights the limits of the complexity of the actions observable in the current benchmarks, e.g. VidSTG.(Zhang et al., 2020).

**Additional results.** We show additional results about the prompt ablation and the impact of the tracking module in Appendix A.2. We add MaskCaptioner qualitative examples in Appendix A.1 and A.5, and discuss the failure cases and limitations of our method in Appendix A.3, as well as give further implementation details in Appendix A.4.

## 5 CONCLUSION

We propose an approach to generate synthetic object-level captions using a state-of-the-art VLM and extend the LVIS and LV-VIS datasets with synthetic captions. We use the resulting LVISCap and LV-VISCap datasets to train MaskCaptioner, a DVOC model that can simultaneously detect, segment, track, and caption objects throughout a video. With finetuning, MaskCaptioner achieves state-of-the-art performance on the VidSTG, VLN and BenSMOT benchmarks, while extending the DVOC task to segmentation masks.

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

## A APPENDIX

In this appendix, we show qualitative results from our method in Section A.1, present additional ablations in Section A.2, discuss failure cases and limitations in Section A.3, share more details about the datasets, the method and the implementation in Section A.4, and show some more qualitative examples in Section A.5.

### A.1 QUALITATIVE RESULTS

In Figure 5, we show qualitative DVOC results from MaskCaptioner on (i) the VidSTG dataset (Zhang et al., 2020) with (box, caption) pairs predictions, and (ii) the LV-VIS dataset (Wang et al., 2023) with (mask, caption) pairs predictions. These examples show that MaskCaptioner has learned to predict captions that focus on the localized objects while integrating high-level scene understanding. More examples are shown in Fig. 8. We note that, when tuned on VidSTG, MaskCaptioner produces less informative and less descriptive captions. This is due to the VidSTG annotation captions being designed for grounding rather than for captioning or DVOC, and thus being only little descriptive, little informative, and overlooking to the global context. In contrast, when trained on LVISCap and LV-VISCap, MaskCaptioner visually generates much richer and accurate descriptions, further highlighting the value of our synthetic captions.

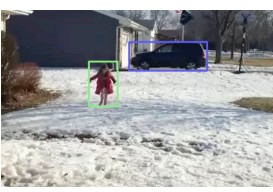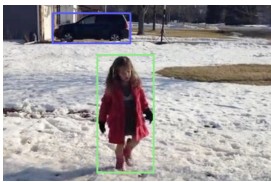

A child in red is in front of a black car.

There is a black car behind a child in red.

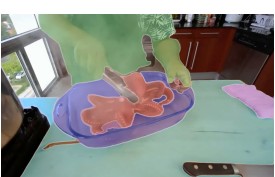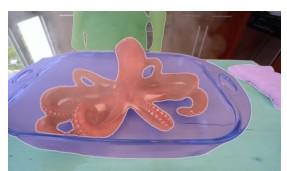

A person wearing an apron is holding an octopus in a glass bowl.

The glass bowl is being filled with an octopus by a person's hand.

The octopus is being placed inside a glass bowl by a person.

The wooden table is supporting a glass bowl filled with an octopus being handled by a person.

A white towel is placed on the wooden table near a knife.

Figure 5: Qualitative examples, obtained with MaskCaptioner on VidSTG (first row, with bounding boxes), and on LV-VISCap (second row, with segmentation masks).

### A.2 ADDITIONAL ABLATIONS

Table 9: **Impact of the prompting strategy on caption quality**. Scores are given by a human evaluator from 0 to 2 (incorrect, partially correct, or correct) on a subset from the LV-VIS validation set, and brought to 0-100 range. For the mask visual prompt experiments, we use our best prompt with either the object's bounding boxes or center point coordinates as a localization cue in the text prompt.

| Visual prompt | Prompting method | Average rating | Rating percentage | | |
|---|---|---|---|---|---|
| | | | 0 | 1 | 2 |
| bounding boxes | single frame | 26.8 | 66.3 | 13.9 | 19.9 |
| | + multiple frames | 27.1 | 68.7 | 8.4 | 22.9 |
| | + detailed instructions | 29.5 | 65.0 | 10.8 | 24.10 |
| | + category labels | 80.7 | 10.8 | 16.9 | 72.3 |
| | + bounding box coordinates | 83.1 | 9.6 | 14.5 | 75.9 |
| | + bounding box area | 84.3 | 7.8 | 15.7 | 76.5 |
| | + few shot examples | **85.1** | 9.1 | 11.5 | 79.4 |
| mask boundaries | center point coordinates | 75.9 | 17.5 | 13.2 | 69.3 |
| | bounding box coordinates | 77.1 | 15.7 | 14.5 | 69.9 |

**Prompting strategy.** In Table 9, we show the distribution of ratings given by the human annotator depending on the prompting strategy. Using the box visual prompt yields a better focus on the queried object and more correct object captions. Importantly, giving the category labels in the prompt helps the model to generate more accurate object captions. Overall, the rate of correct captions with the best prompt indicates good quality for our synthetic object captions.

**Tracking module.** In Table 10, we show that the top-k enhanced tracking (Zhu et al., 2024) is important for tracking objects effectively on the challenging VidSTG dataset (Zhang et al., 2020), as seen in the AssA, CapA and CHOTA scores. We attribute this difference to the numerous objects that disappear for a significant number of frames in the long videos of VidSTG. The top-K approach uses a memory bank of tracked queries that helps keeping track of these entities, while they are lost using the $i$ to $i + 1$ tracking from OVFormer (Fang et al., 2025).

Table 10: **Impact of the tracking module on VidSTG DVOC.**

| Tracking method | CapA | DetA | AssA | CHOTA |
|---|---|---|---|---|
| OVFormer module (Fang et al., 2025) | 51.9 | **67.0** | 58.2 | 58.7 |
| Top-K enhanced module (Zhu et al., 2024) | **52.7** | 66.8 | **71.0** | **63.0** |

## A.3 FAILURE CASES AND LIMITATIONS

### A.3.1 FAILURE CASES

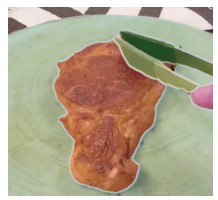 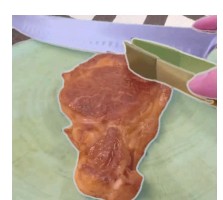

The knife is being used to cut a steak on a wooden cutting board.

The steak is being cut with a knife on a wooden cutting board.

A person is holding a steak with tongs on a wooden cutting board.

The **knife** is being held by a person next to a steak on a wooden cutting board.

The wooden chopping board holds a steak being cut with a knife and a pair of tongs.

(i) Recognition error

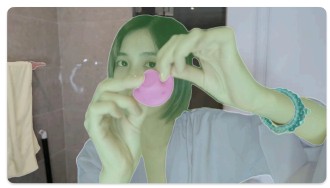 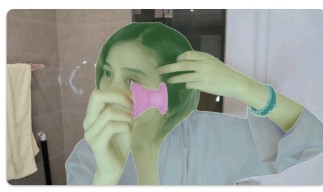 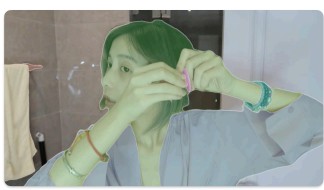

A person is holding a roll of tape in front of them. The bracelet is worn on the wrist of a person who is holding a pink sphygmomanometer. The yellow robe is being worn by a person who is holding a pink spinner. The earring is being help up by a person in front of a mirror. The towel rack is mounted on a wall in a bathroom. A white bath towel hangs on the wall.

(ii) Inconsistent object categories

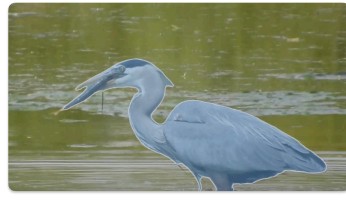 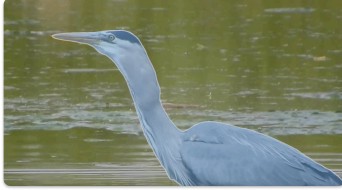

The gray heron is standing in the water with a fish in its beak.

(ii) Detection/segmentation error

Figure 6: Some DVOC failure cases of MaskCaptioner observed on the LV-VIS dataset.

We observe 3 main types of failure cases for our approach and illustrate them in Figure 6:

**(i) Recognition error**: in the case of ambiguous context, blurred instance or rare categories, MaskCaptioner might fail to recognize the object it is describing, sometimes leading to a wrong denomination (e.g. here, a rare "pair of tongs" is incorrectly denominated as "knife"). **(ii) Inconsistent captions** : in similar situations, the captions produced by MaskCaptioner can be inconsistent when referring

to the same object in different captions. **(iii) Detection/segmentation error** : In case of complex movement, appearance change or occlusion, MaskCaptioner sometimes fails to detect, segment, or track objects, leading to missing captions (e.g. here, the fish is not detected in the beak of the heron, and thus has no associated caption).

### A.3.2 Limitations

**Limitations.** While our approach outperforms the state-of-the-art in dense video object captioning, there is still room for improving localization and captioning. Localization sometimes fails, in particular for small objects. Furthermore, the automatically generated captions are, in some cases, too generic, and can mix up two objects of the same class in the video. Future work could investigate different automatic captioning techniques for DVOC, for example based on an approach such as GLaMM (Rasheed et al., 2024), which generates captions with grounding associated to the mentioned objects. Eventually, objects in the videos often perform a single or few actions, and we believe that it is important for future works to build benchmarks with more complex object interactions, for instance with multiple action segments.

### A.4 Additional details

#### A.4.1 Prompting strategy details

The full prompt template used to generate our LVISCap and LV-VISCap datasets is illustrated in Figure 7. For a video $x$ with $N$ objects, we prompt the VLM $N$ times, and for each object $j$ the prompt is composed in three parts: (i) the static system prompt $p_s$ gives general instructions for object-level caption generation, practical rules, prompting format and an example, (ii) the user prompt $p_u(j)$ depends on the example and contains textual annotation information to help the model describe objects and interactions accurately. These information notably contain target object positions, areas, category, and the categories of other objects in the scene, (iii) the visual prompt $\hat{x}_j$ consists of 4 sampled frames with drawn bounding boxes for object $j$.

#### A.4.2 Dataset details

**LVIS** (Gupta et al., 2019a) is a large-vocabulary image instance segmentation dataset with a long-tail distribution of 1203 annotated categories, for a total of over 2.2 million annotations in 164k natural images. The dataset is split in a training set with 100k images and 1.2M annotations, a validation set with 19k images and 244k annotations, and two test sets with 19k images each.

**LV-VIS** (Wang et al., 2023) is a recent large-vocabulary video instance segmentation (VIS) benchmark. It comprises $4,828$ videos with over 26k video segmentation masks from 1,196 object categories, with an average of over 5.4 objects per video. The data is split into a training set of $3,076$ videos and 16k video-level annotations, a validation set of 837 videos and 3.7k annotations, and a test set with 908 videos.

**LVISCap** and **LV-VISCap** denote our extensions of LVIS and LV-VIS (see Section 3.1). LVISCap extends LVIS with a total of $1,488,354$ synthetic captions, including $1,244,271$ training annotations and $244,083$ validation annotations. LV-VISCap includes a total of $19,717$ synthetic captions for $16,017$ training and $3,700$ validation annotations. Note that in the absence of annotations on the test sets of LVIS and LV-VIS, we only extend the training and validation sets with captions, and use the validation set for evaluation.

**VidSTG**(Zhang et al., 2020) is a spatio-temporal video grounding dataset, containing $6,924$ videos with $44,808$ exhaustive trajectories annotations over 80 categories, as well as object sentence descriptions (for some objects and some timestamps only), which serve as queries for grounding. Zhou et al. (2023) repurposed the dataset for DVOC by using queries as captions, and by excluding annotations without captions during evaluation. Following Zhou et al. (2023), we sample 200 frames uniformly across each video for both training and evaluation. Overall, the repurposed VidSTG training set counts $28,275$ object trajectories, with $15,182$ object captions. The validation set, used for DVOC evaluation, includes 602 videos with $1,644$ captions.

**Video Localized Narratives (VLN)** extends existing datasets with "narrations of actors actions" in videos. We use the subset from the UVO dataset, which contains 3 sparsely annotated frames with non exhaustive captions for a total of $5,136$ training and $2,451$ validation videos.

### System Prompt $p_s$

```
"""
Generate a caption for a video, focusing on a queried object highlighted with green bounding boxes, and
semantic class provided.
It should be a rich sentence describing the object's APPEARANCE and ACTION, trajectory, or interaction with
other objects in the video frames. The other objects are not highlighted and should be mentioned only if
they interact with the query object or are relevant to the context.

# Rules
    - The single query object highlighted with bounding boxes in the frames SHOULD be the subject of the
    sentence. ex: for category "bottle": "The bottle is being inspected by a person"
    - Only facts that are visible in the video should be mentioned.
    - You should NOT mention the fact that the query object is highlighted with green bounding boxes.
    - You should RETURN A CAPTION no matter what, even if the query object is not visible in any frame.
    - If multiple objects of the same class as the query object are visible, the caption should focus
    exclusively on the single highlighted object and describe it as the singular subject of the sentence.
    - No foreign alphabets or special characters should be used in the caption. Translate foreign words if
    needed.

# Input Details
    - **Frames**: Provided sequence of 4 frames sampled from a video, in which a bounding-box highlights a
    query object
    - **Bounding Box**: Locations of the query object in the respective frames, in the format [(xmin, ymin,
    xmax, ymax),...] with each value ranging from 0 to 1000 representing a percentage of image dimensions.
    - **Area**: Area of the query object in the image, as a percentage of the total image area. This could
    help to determine wether the object is in the background. (big object class with small area)
    - **Semantic Class**: Class of the query object to be described in the caption
    - **Other classes**: Some classes of other objects in the video, These objects are not highlighted and
    should be mentioned only if relevant.

# Examples
    - **Input**: An image showing a women dressed in black and white and a dog both running on a beach with
    people in the background. The woman's short is highlighted with green bounding boxes in the video. object
    class: "short pants".
    - **Output**: "A black and white short pants is being worn by a woman running with a dog across the sandy
    beach"
"""
```

### User Prompt $p_u(j)$

```
"""
Query object Bounding Box location in the respective
frames: {formatted_normalized_bbox}
Query object areas in percentage of the respective frames :
{formatted_area_pct}
Query object Class: '{cat["name"]}'
Other classes: {", ".join(other_cat_names)}
Generate a one-sentence caption, focusing on the object of
class '{cat["name"]}' highlighted by the bounding boxes.
"""
```

### Visual prompt $\hat{x}_j$

$\hat{x}_j^{i_1}$ ... $\hat{x}_j^{i_4}$

Sampled frames with drawn bounding boxes $\mathcal{B}_j^{i_1}$ ... $\mathcal{B}_j^{i_4}$

Figure 7: Prompt template used to generate synthetic object captions from video segmentation annotations for the LV-VIS dataset Wang et al. (2023). The system prompt $p_s$ contains general instructions, while user prompt and visual prompt $p_u(j)$ and $\hat{x}_j$ are formatted with information from each annotation.

**BenSMOT** contains manually collected annotations of bounding box trajectories and associated captions, focusing exclusively on humans in videos. It includes an average of 2.2 instances per video, and counts $2,284$ videos for training and $1,008$ for evaluation.

### A.4.3 MORE IMPLEMENTATION DETAILS

The visual backbone is initialized with weights pretrained on ImageNet-21K Deng et al. (2009) following OVFormer Fang et al. (2025), and the Mask2Former Cheng et al. (2022) weights are trained from scratch. The OVFormer classifier uses a frozen CLIP ViT-B/32 Radford et al. (2021) encoder. The captioning head is initialized with weights from BLIP-2 Li et al. (2023a) with frozen LLM OPT-2.7B Zhang et al. (2022a) following Chouduri et al. Choudhuri et al. (2024).

For all experiments except LV-VIS tuning, we first train the segmentation/detection model, then freeze it and tune the captioning head. Respectively for LVIS/VidSTG/LV-VIS we train for $440k/40k/22k$ for the first stage and $5k/2k/2k$ for the second stage. When tuning pretrained models on Vid-STG/VLN/BenSMOT, we train the two stages for $(15k, 2k)/(15k, 500)/(15k, 2k)$ steps, whereas for LV-VIS, we end-to-end tune the model for $2k$ steps. Experiments are run with a batch size of 8, except when using LVIS+COCO and LV-VIS where we use a batch size of 4 and for video-level tuning of the captioning head where we use a batch-size of 1. Experiments on LV-VIS are end-to-end trainings with clip-level supervision only. For VidSTG and BenSMOT experiments we use video-level tuning

for captioning with temporal aggregation, with $T_{\text{agg}} = 8$. For all experiments we train the model with a clips of size $T = 2$, and at inference use $T = 5/1/1/1$, $T_{\text{match}} = 1/100/20/40$, $K_{\text{match}} = 1/7/5/7$ for LV-VIS/VidSTG/VLN/BenSMOT experiments respectively. For the largest dataset (COCO + LVIS) the optimization takes 2 days on 4 H100 GPUs.

Following OVFormer Fang et al. (2025), for all datasets we use an AdamW optimizer and the step learning rate schedule, with an initial learning rate of 0.0001 and a weight decay of 0.05, and apply a 0.1 learning rate multiplier to the backbone. We decay the learning rate at 0.9 and 0.95 fractions of the total number of training steps by a factor of 10. For respectively image/video datasets, we resize the shortest edge of the image to 800/480 for SwinB and 800/360 for ResNet for training and inference.

### A.5 More qualitative examples

More successful qualitative results obtained with MaskCaptioner are presented in Figure 8, complementing Figure 5. MaskCaptioner effectively learns to jointly segment, detect, track and caption object trajectories.

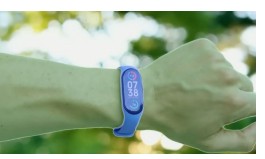 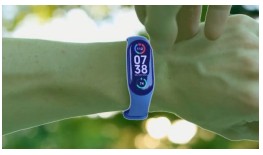

A person is wearing a blue smartwatch with a heart rate monitor on their wrist.

The light blue smartwatch displays the time on its face as it rests on a persons wrist.

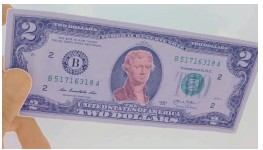 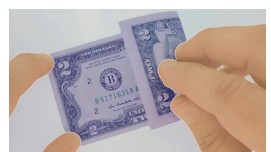

A person is holding a small dollar bill.

The two dollar bill is being held in the hand of a person.

A person is featured on the two-dollar bill.

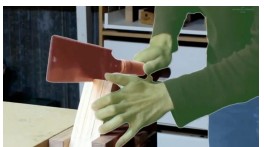 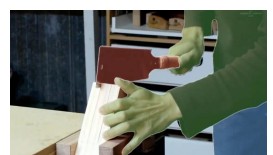

The person is using a handsaw to cut a piece of wood.

The jean is being worn by a person working on a piece of wood with a handsaw.

The handsaw is being used by a person to cut a piece of wood.

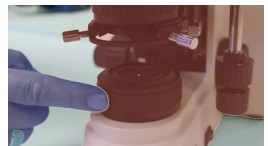 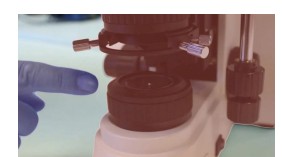

A person is seen adjusting the focus of a microscope.

The microscope is being adjusted with a person's hand.

A ring is worn on the finger of a person who is examining a microscope.

(i) LV-VIS examples

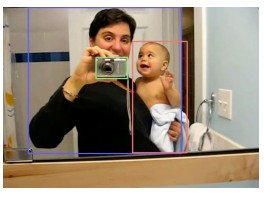 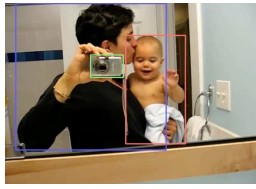

A baby leans on an adult in the bathroom.

An adult in black clothes holds a baby.

There is a camera in front of an adult.

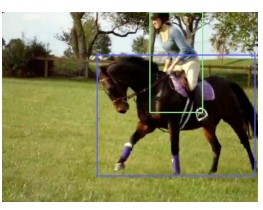 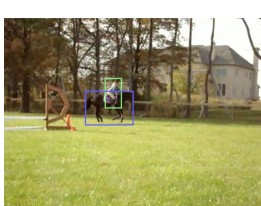

An adult rides the horse on the grass.

There is a horse towards an adult on the grass.

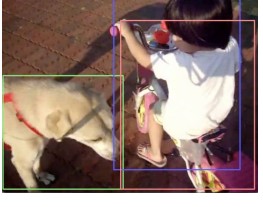 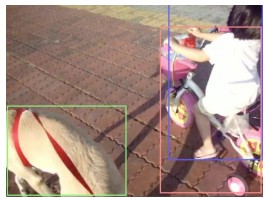

There is a dog away a bicycle on the ground.

A child in white is above a pink bicycle.

There is a pink bicycle beneath a kid on the ground.

(ii) VidSTG examples

Figure 8: More successful DVOC qualitative examples obtained with MaskCaptioner on (i) LV-VIS (with segmentation masks) and (ii) VidSTG (with bounding boxes).

