# OpenReview forum: "MaskCaptioner: Learning to jointly segment and caption object trajectories in videos"
_ICLR.cc/2026/Conference — ICLR 2026 Conference Withdrawn Submission_

### Official Review · Reviewer_Fsiv · 2025-10-21

**Soundness:** 2
**Presentation:** 2
**Contribution:** 2
**Rating:** 2
**Confidence:** 5

**Summary:**

This paper proposes to generate spatio-temporally localized entity captions with VLM for the task of dense video object captioning, thereby jointly detecting, segmenting, tracking and captioning object trajectories.

**Strengths:**

S1. The paper is well organized and easy to follow.
S2. The paper extends existing datasets with synthetic data of object captions, boxes, and segmentation masks.

**Weaknesses:**

W1. Lack of Technical and Theoretical Novelty. The main contribution of this paper is a data generation pipeline that heavily relies on existing VLM models. This process is more like a prompt engineering architecture, where the customized design of the proposed method is trivial. Additionally, the proposed MaskCaptioner framework is an explicit combination of existing OVFormer and BLIP-2 models. The technical and theoretical contributions of this paper need further clarification.

W2. Insufficient Experimental Validations. The latest comparison method is published in 2024, how about comparing with more recent methods (e.g. OVFormer)? Additionally, the proposed method adopts SwinB and ResNet50 backbones, could it applies to VLM-based methods or compare with VLM-based methods?

**Questions:**

Q1. The technical and theoretical contributions of this paper need further clarification.
Q2. The author should include comparisons with more recent (2025) methods to demonstrate the effectiveness of MaskCaptioner further.
Q3. How about applying MaskCaptioner beyond SwinB / ResNet50 backbones, such as VLM models?
Q4. How about comparing MaskCaptioner with VLM-based dense video object captioning methods?

---

### Official Review · Reviewer_msHc · 2025-10-30

**Soundness:** 2
**Presentation:** 3
**Contribution:** 3
**Rating:** 4
**Confidence:** 3

**Summary:**

Focusing on the task of Dense Video Object description (DVOC), the authors propose to use advanced visual language models to generate synthetic descriptions of spatio-temporal localized entities, which are extended to obtain the LVISCap and LV-VISCap datasets containing mask, bounding box, category, and description annotations. Based on this, we design and train MaskCaptioner, an end-to-end model that can jointly complete the tasks of object detection, segmentation, tracking, and trajectory description, achieving SOTA performance on three DVOC benchmark datasets.

**Strengths:**

1.	Authors first propose a VLM based object-based synthetic description generation method for video, which is an effective method to solve the high cost of fine annotation in traditional DVOC tasks.
2.	This paper constructs the MaskCaptioner end-to-end framework, achieving the first deep joint optimization of four major tasks in video, segmentation, detection, tracking, and captioning, making it practically valuable.
3.	MaskCaptioner achieves state-of-the-art on three DVOC benchmark datasets VidSTG, VLN, and BenSMOT.

**Weaknesses:**

1.	The description of some experimental scenarios is not clear. Tables 4 and 5 only illustrate the fine-tuning dataset and do not mention which pre-training data was used. Is it possible to compare the effects under different pre-training data as in Table 3, so as to verify the effectiveness of the generated dataset.
2.	Some modules of MaskCaptioner (such as Top-K tracking and time aggregation) are verified to be reasonable through comparative experiments, but there are still some areas to be improved. For example, the two-stage training strategy can be compared with the two-stage training strategy to verify whether the strategy is the optimal solution of the performance-efficiency trade-off. And to verify the synergy between the segmentation-tracking-description modules.
3.	The DVOC comparison method can be suitably enriched. In addition, in OW-VisCapTor in Table 1, the authors only compare their COCO pre-training results on VidSTG, but not their fine-tuning performance on LV-VISCap, which cannot fully reflect the advantages of MaskCaptioner in segmentation + description tasks.

**Questions:**

Will you consider validating the performance of synthetic description generation strategy and MaskCaptioner on public video datasets such as Kinetics-400 and ActivityNet? In this way, the effectiveness of the generation strategy can be better verified.

---

### Official Review · Reviewer_VhTQ · 2025-10-31

**Soundness:** 2
**Presentation:** 2
**Contribution:** 1
**Rating:** 2
**Confidence:** 4

**Summary:**

This paper introduces MaskCaptioner, a framework addressing Dense Video Object Captioning (DVOC). DVOC is a complex task that requires the joint execution of object detection, tracking, segmentation, and natural language captioning for object trajectories within a video. The authors propose generating captions using VLM to extend existing datasets for training. They then present an end-to-end trainable approach designed specifically for this combined task, claiming to move beyond suboptimal disjoint training strategies. The paper reports superior performance across various datasets, suggesting the value of the proposed methodology and the synthetic data generation approach.

**Strengths:**

1. Dataset Augmentation: The strategy of employing a state-of-the-art VLM with targeted prompts to generate spatio-temporal captions is a resourceful way to address the problem of limited, high-cost, human-annotated DVOC training data;
2. Unified Methodology: The proposal of an end-to-end trainable model MaskCaptioner that simultaneously handles segmentation, tracking, and captioning is conceptually robust.

**Weaknesses:**

1. The fundamental significance of the research question is unclear and not sufficiently justified. The paper fails to articulate the concrete value proposition of the DVOC task, particularly its advantage over directly using powerful VLMs.
(1) Redundancy and Granularity: As shown in the figures like Figure 1, the captions (e.g., for "human" and "sharpener") appears highly redundant and the tracking of static entities (like the "pencil") may seem visually meaningless, questioning the necessity of adding tracking and dense segmentation to the overall goal of description. And using VLM could get better description for the given videos.
(2) Segmentation Justification: The paper does not convincingly demonstrate why the introduction of the segmentation task is essential or beneficial for producing superior captions compared to using only robust detection and tracking bounding boxes (PVOC). A clear comparison with a non-segmentation baseline is needed.
2. Confusing writing and content organization. In the abstract, the authors claim that previous methods on PVOC leverage disjoint training strategies thus leading suboptimal performance. But I read the reference papers and found that $SMOTer^{[1]}$ is an end-to-end method for PVOC with proposed human-annotated dataset. And for the drawbacks of disjoint training that is emphasized, I think you should conduct some ablation studies on your proposed architecture, MaskCaptioner, to illustrate the better performance of joint training compared to disjoint training.

 [1] Beyond mot: Semantic multi-object tracking. ECCV 2024.

**Questions:**

Please see the questions listed in Weaknesses part.

---

### Official Review · Reviewer_qzek · 2025-11-01

**Soundness:** 3
**Presentation:** 3
**Contribution:** 3
**Rating:** 4
**Confidence:** 5

**Summary:**

This paper introduces MaskCaptioner, the first end-to-end model that detects, segments, tracks, and captions object trajectories in video. By extending Dense Video Object Captioning (DVOC) from bounding boxes to masks, it unifies vision-language understanding at the spatio-temporal level. To address the scarcity of object-level video-caption pairs, the authors generate rich synthetic captions for two large segmentation datasets (LVIS and LV-VIS) using a multimodal prompting pipeline, creating new resources—LVISCap and LV-VISCap—that include masks, boxes, categories, and captions. Extensive experiments show that MaskCaptioner achieves state-of-the-art performance on three DVOC benchmarks (VidSTG, VLN, and BenSMOT), supported by ablation studies, qualitative examples, and error analysis.

**Strengths:**

1. The paper is first to demonstrate joint segmentation, tracking, and captioning of object trajectories at the mask level in videos, unifying previously fragmented pipelines. The full MaskCaptioner architecture (Fig. 3) is clearly presented and technically solid.
2. Results in Table 3 show MaskCaptioner achieving higher CHOTA and CapA scores than competing methods, with explicit gains in captioning components when pretraining on synthetic caption datasets.
3. This paper is easy to follow.

**Weaknesses:**

1. While the system is well-integrated and the dataset generation pipeline is carefully engineered, the core architectural contributions are largely incremental, building on existing modular advances (e.g., Mask2Former, OVFormer, BLIP-2, top-K tracking from Zhu et al., 2024). The main novelty lies in the aggregation and combination of components rather than fundamental changes to underlying modules. Architectural modifications, such as extending detection-classification heads with captioning modules, follow expected design patterns.
2. Despite introducing new synthetic datasets, benchmark evaluations rely on either synthetic captions or repurposed datasets (VidSTG, VLN) whose natural language supervision is sparse, ambiguous, or not specifically designed for DVOC. As noted in Figure 5 and the appendix, VidSTG captions are intentionally less informative, which leads to less descriptive outputs from MaskCaptioner. The strongest seg+caption results are self-evaluated on synthetic test splits, which may have limited diversity or language richness. The performance gap between manually-annotated versus synthetic test data, highlighted in Table 6, appears under-explored.
3. The quality of LVISCap and LV-VISCap depends entirely on the VLM used for caption generation. Any errors, biases, or limitations in the VLM’s descriptions of objects or actions will propagate into the training set and ultimately affect MaskCaptioner’s performance.
4. Although the synthetic data is high-quality, it is derived from existing datasets (LVIS, LV-VIS). Consequently, the model’s ability to generalize to completely novel or unconstrained videos may be limited by the domain of the source datasets used for VLM prompting.
5. While MaskCaptioner outputs both masks and boxes (Fig. 8, Table 4), the relationship between segmentation quality and captioning accuracy is not fully explored. It remains unclear whether high-quality masks always lead to better captions, or if there are trade-offs. Current analysis largely treats detection, tracking, and captioning metrics independently, with limited insight into cross-task dependencies or joint failure modes. More in-depth evaluation of interactions between tasks would be valuable.
6. The contribution is primarily empirical and engineering-driven. The work does not introduce new theoretical or algorithmic insights into learning, convergence, or the limits of mask-based video captioning.

**Questions:**

1. Table 6 highlights performance differences between synthetic and manually annotated evaluations. Do the authors expect that scaling to even larger synthetic datasets would improve generalization, or might it further amplify annotation bias? What strategies could be employed to mitigate potential biases?
2. Have the authors considered or explored incorporating object relationships or interactions into the captioning head or annotation pipeline—for example, describing pairs or triplets of objects and their interactions? What challenges or potential benefits did they observe in this context?

---

### Note · Authors · 2025-11-14

I have read and agree with the venue's withdrawal policy on behalf of myself and my co-authors.